

# Scenario-feature identification from online reviews based on BERT

Xunjiang Huang and Kang Yan

School of Business Administration, Northeastern University, Shenyang, Liaoning Province, China

## ABSTRACT

Scenario endows a product with meanings. It has become the key to win the competition to design a product according to specific usage scene. Traditional scenario identification and product feature association methods have disadvantages such as subjectivity, high cost, coarse granularity, and limited scenario can be identified. In this regard, we propose a BERT-based scenario-feature identification model to effectively extract the information about users' experience and usage scene from online reviews. First, the scenario-feature identification framework is proposed to depict the whole identification process. Then, the BERT-based scene-sentence recognition model is constructed. The Skip-gram and word vector similarity methods are used to construct the scene and feature lexicon. Finally, the triad is constructed through the analysis of scene-feature co-occurrence matrix, which realizes the association of scenario and product features. This proposed model is of great practical value for product developers to better understand customer's requirements in specific scenarios. The experiments of scenario-feature identification from the reviews of Pacific Auto verifies the effectiveness of this method.

## INTRODUCTION

Users actively give feedback about shopping feelings and usage experiences through the evaluation function of shopping platforms (*Hu, Liu & Zhang, 2008*; *Bian et al., 2022*), which generate massive online reviews. The large amount of online review data contains rich valuable information such as customer's usage scenes and experience. That information provides direction for designers to improve and refine their products. Nowadays, users not only need functional satisfaction but also pursue the achievement of immersive scenario experience (*Chitcharoen et al., 2013*). It is the key to obtain users that product can provide users with multi-scene usage experience in their consuming process of products (*Yang & Huang, 2015*; *Benford et al., 2009*). The car is not only a traditional transportation tool, but also a comprehensive usage scenario of "transportation + entertainment + leisure". Bookstores and cafes are gradually shifting from a single service provider to a lifestyle leader that integrates multiple scenarios. There is a growing tendency among manufacturers to configure products with different functional features according to different usage scenarios. The scenario-based product development provides more innovative portfolios, which is closer to the real needs of users. *Lu & Liao (2023)* combined experience design and scenario-based design approaches to extend the interaction system

Corresponding author
Xunjiang Huang,
xjhuang@mail.neu.edu.cn

of the home cleaning robot to enhance the experience value of the product. *Asgarpour et al. (2023)* proposed an integrated scenario-based model that defines four possible future scenarios to assess possible changes in road infrastructure and provide recommendations for investment in infrastructure. Online reviews provide information about the experience of consumers on each dimension of product features, and contain information about the scenarios how consumers use products. Scenario is the connector between technology and demand, and users have different requirements in different scenarios. In a specific scenario, user needs actively seek professional experience, and manufacturers can better grasp user habits and pain points through usage scenarios analysis. Then they can improve products with the features that users are more concerned about in given scenario, which strengthen their competitiveness by increasing the degree of product heterogeneity. The idea of product as scene has become a deep-rooted theme (*Suri & Marsh, 2000*). Users can more easily select and define their product requirements only when the product is placed in a specific scene. The identification of scenario is of great value to customer needs (*Hickey, Dean & Nunamaker, 1999*).

"Scenarios give meanings to product" (*Suri & Marsh, 2000*), and scenario-based design can more effectively meet user requirements in specific contexts and effectively improve user satisfaction (*Li et al., 2021*). In order to enhance the educational game experience and short-term learning, *Xinogalos & Eleftheriadis (2023)* relate the game experience with realistic technology work scenarios to explore the impact of games on players However, there are relatively few studies on scenario-related topics, and there is a lack of research on scenario construction and identification. *Zhao, Peng & Wang (2020)* proposed a scenario-based user experience design evaluation method about the user experience of smart home products to determine product design solutions by five steps, which includes human determination of product usage scenarios to improve user satisfaction and product quality. Studies on scene identification are mostly based on methods such as expert discussions, which first analyze the functional characteristics of products, then analyze them for clustering, and then classify product usage scenarios. *Mhaidli & Schaub (2021)* adopted the scenario construction method to identify the use of manipulative techniques in advertising. Such classification and identification methods are mostly based on subjective experience, which is difficult to adapt to the personalized scene needs of large-scale user groups (*Van Helvert & Fowler, 2003*; *Zhang & Wang, 2022*). In this article, we attempt to establish a user scenario identification model by using BERT, Skip-gram model and other data mining methods to extract usage scenarios from online reviews. And then the product scenario-feature association model is built to match them.

## RELATED WORK

Scenarios, which first referred to images in artworks such as movies or operas, were applied to product design and development in the 1990s, and the benefit-cost ratio to design features by analyzing user scenarios was significantly higher than that of to improve features by testing (*Holbrook, 1990*). In the era of mobile connectivity, scenarios were regarded as the stories between people and their activities, and scenario-based design can improve the usefulness of products (*Li et al., 2021*). Today, product design more centers on

users and focuses on enhancing their usage experience. Consumers are no longer satisfied with the functional utility of a product, but pursue a comprehensive cross-sensory experience by taking the product as a carrier (*Dong & Liu, 2017*). Scenario-based products are solutions with user experience as the core (*Suri & Marsh, 2000*; *Anggreeni, 2010*), and users' needs and habits in various scenarios are the basis for their purchasing behaviors. Effectively identifying and satisfying users' personalized needs in different scenarios is an imminent and prominent problem faced by manufacturers in product development (*Yang & Huang, 2015*).

*Van Helvert & Fowler (2003)* proposed the Scenario-based User Needs Analysis (SUNA) method for scenario identification, in which a certain number of experts first formed a seminar group, each expert conceived scenarios and analyzed them separately, and then the scenario analysis results of all experts were integrated and processed. *Suri & Marsh (2000)* argue that scenarios complement anthropic factors and are a powerful tool in the early stages of product design, and they describe some of the advantages and potential pitfalls in using scenarios and provide examples of how and where to apply them effectively. *Anggreeni & van der Voort (2007)* proposed a classification framework for creating and applying different types of scenarios in the product design process to better understand the role that scenarios played in the product design process. The product lifecycle-based approach places more emphasis on systematic requirements, and scenarios both encapsulate and derive requirements. *Dong & Liu (2017)* argue that there are complex interactions among users, products, and environments. The product usage scenarios are the basis for the formation of user experience which cannot occur without the role of scenarios. Therefore, scenes are functions of time, environment, users, products, and behaviors, and are categorized according to them. The three-dimensional matrix of F-E-S association mapping is adopted to associate features and scenes. The evaluation indexes of user experience in each scene are calculated iteratively by constructing the association matrix of incremental mapping. *Yang et al. (2022)* analyze the relationship between multi-group requirements and products in order to improve the rationality of a product. They also analyze the constituent units of scenarios and the mapping relationship between scenarios and functions. Then, the scenarios and user requirements are translated into interrelated functional features of products. *Li et al. (2021)* proposed an identification method of innovation opportunities for Internet products by combining scenario analysis and the Kano model to evaluate key requirements, in which hierarchical analysis models are adopted to select useful scenario concepts, a method that has been effectively tested in practical software innovation. *Ouyang & He (2018)* constructed a scenario-based designing method relevant to user experience, dividing the user experience in the given scenarios into three parts, namely user goals, behavioral patterns, and user scenarios. And the application in the interior lighting function of a car verified its rationality and practicality.

With the advent of the experience economy, online reviews influence consumers' purchase attitudes and purchase decisions (*Weathers, Swain & Grover, 2015*). They are increasingly becoming an important information source for manufacturers' designing because they contain the information about consumption scenarios and usage experiences. The concept of online reviews, first formally introduced by *Chatterjee (2001)*, refers to the

unstructured comments and recommendations on various aspects of products generated by users during the usage after purchasing related products or services online or offline (*Chen & Xie, 2008*), which is a kind of word-of-mouth information in the Internet era, also known as online consumer reviews (*Mellinas, Nicolau & Park, 2019*). Compared with traditional data acquisition methods such as questionnaires and interviews, online reviews are easy to collect, less costly, and independent of survey respondents. They are more realistic and objective, and can cover numerous detailed features of products. Although their composition takes many forms, they are still quite valuable and representative after processed properly (*Xiao, Wei & Dong, 2016*; *Lycett, 2013*). Current researches on online reviews mainly focus on sentiment analysis (*Kumar et al., 2021*; *Colón-Ruiz & Segura-Bedmar, 2020*), user recommendations and purchase decisions (*Willemsen et al., 2015*; *Yang, Cheng & Tong, 2015*), feature identification, usefulness (*Chatterjee, 2001*; *Majumder, Gupta & Paul, 2022*) and their impact on product pricing and sales, *etc*. *Jensen et al. (2013)* introduced persuasion models into the analysis of online reviews to explore how they affect consumers' perceptions of products. *Huang et al. (2013)* analyzed the influence of product reviews on purchase decision behavior based on cognitive matching (*Vessey & Galletta, 1991*) and graphical consistency (*Mandler & Parker, 1976*) by classifying product reviews into two forms about features and experiences. By collecting users' online reviews on Taobao platform, *Mo, Li & Fan (2015)* explored the intervention mechanism of product reviews on consumers' purchase behavior based on S-O-R model from the perspective of consumer learning. *Yu, Debo & Kapuscinski, 2016* explored the role of online reviews in assistant decision making about the pricing of a firm's products. *Zhu et al. (2019)* explored the impact of discounts on online reviews, and found that users who received discounts rated them higher, while those who did not receive had greater diversity of expression and expressed a more comprehensive range of content. However, *Li & Hitt (2010)* argued that price also inversely affected consumers' evaluation on products. *Wei et al. (2010)* constructed a semantic-based product feature extraction method from online reviews, and experiments showed its effectiveness. *Yan et al. (2015)* proposed a new type of feature extraction method by integrating extended PageRank algorithm, synonym expansion and implicit feature inference, which can automatically extract product features from online reviews. Although many scholars have analyzed the important influence of online reviews on consumers' purchasing behavior and enterprises' product development from different perspectives, few studies have addressed the mining of information about the usage scenarios and corresponding product feature sentiment preferences embedded in online reviews.

Following the era of big data, the era of scenarios is coming (*Scoble & Israel, 2014*). Scenarios have great potential in addressing user needs It is less costly to analyze user needs in specific scenarios for product design, the key of which lies in effective user participation (*Holbrook, 1990*). User-involved product design improves product quality, improves user satisfaction, increases competitiveness, and thus increases corporate profits (*Ahmad & Qadir, 2022*). It also can improve availability, ease of use, and brand loyalty (*Kim & Jeong, 2023*).The online review is the effective practice of user participation, and where users express their real experience. Traditional scene identification and classification based on

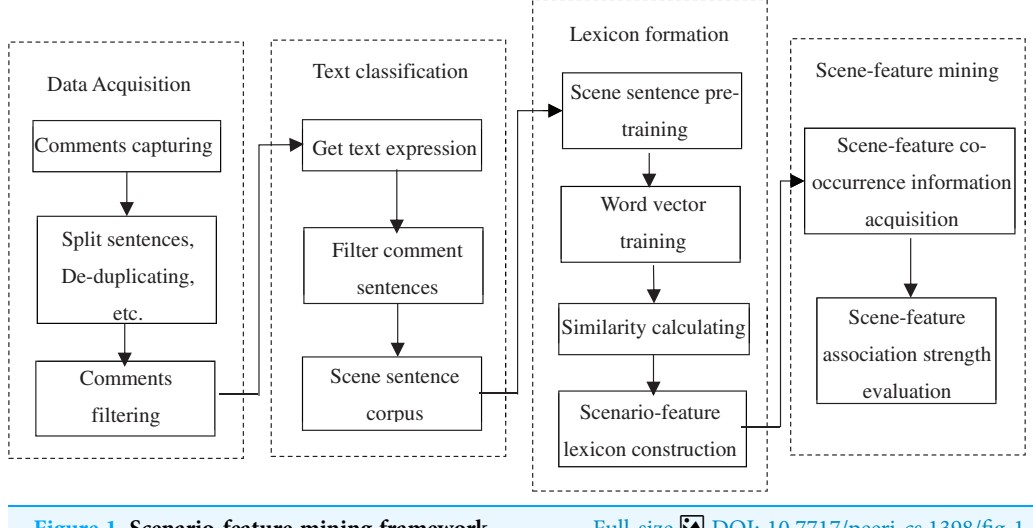

**Figure 1  Scenario-feature mining framework.**

expert judgment are costly, inefficiency and subjective, especially when experts lack relevant domain knowledge, which easily leads to the loss of objectivity (*Li et al., 2020*). And its analysis granularity is coarse, so it is difficult to identify the highly personalized needs in fine-grained scenes. The mining of fine-grained scenarios and user needs from massive online reviews provides guidelines for users' product selection and manufacturer's product innovation. Based on this, this article uses word embedding and other techniques to build a product feature lexicon, obtains the deep semantic representation of online reviews using BERT model, and connects the SoftMax layer to realize the scene description text classification of online reviews. Word co-occurrence is one of the natural features of Chinese language, and the complex network formed by co-occurrence contains rich semantic information such as word similarity and association (*Liu & Cong, 2013*; *Wartena, Brussee & Slakhorst, 2010*), so we combine word co-occurrence and mutual information to realize the association of product features and usage scenes in online reviews. Compared to manual identification of divisions, the use of machine learning can process large amounts of text data faster and can mine fine-grained scenarios and user requirements from massive online reviews, providing guidance for users' product selection and manufacturers' product innovation (*Chen, Wang & Jia, 2023*; *Pan et al., 2022*).

## SCENE MINING MODEL

According to the language structure characteristics of online reviews, which containing the information about usage scenarios, product features and experience preferences, this article proposes a BERT-based scenario-feature mining framework model from online reviews. This framework mainly consists of four modules: data acquisition, BERT-based classifier construction of scenario sentence, scenario and feature lexicon formation, and scenario-feature mining, as shown in Fig. 1.

(1) Data acquisition layer, which crawls reviews, tokenizes sentence and de-duplicates; (2) BERT-based classifier construction of scene sentence layer, which converts the processed comment data into a vector containing semantic features by BERT and utilizes

**Table 1 Examples of online reviews.**

| Comment sentence | Scene | Category |
|---|---|---|
| 车身稳定, 高速下不飘, 转弯也很给力, 没有松散的感觉。(The bodywork is stable, no drift at high speed, and the turning is also very awesome, no loose feeling.) | 转弯 (Turn) | 1 |
| 车身是最好的说明, 因为车的第一眼就是看到车的外观部分, 外观有个性。(The bodywork is the best evidence because the first glance of this car is its distinctive exterior.) | / | 0 |
| 中控有大屏, 都很好用, 悬挂, 发动机都比较满意的了。(The central control has a large screen in good availability, the suspension and engine are both quite satisfactory.) | / | 0 |
| 内饰除了塑料特别多, 其他都还好, 设计也满意 (The interior is fine except for too many plastics used, and the design is also satisfactory.) | / | 0 |
| 隔音很好, 就是悬架有些硬, 座椅舒适度都很好, 手能触摸到的地方都很舒服, 做工精细。(Sound insulation is very good. The suspension is a little hard. The seats are very comfortable. It's very comfortable wherever you can touch it. The whole car has fine workmanship.) | / | 0 |
| 动力说来就来, 没有迟钝, 超车自信很多。(Power comes too soon, no any sluggishness, and there is confidence in overtaking in overtaking.) | 超车 (Overtaking) | 1 |
| 内饰还可以比较满意 看着很有感觉手感都还不错。(The interior is satisfactory and looks to have a good feel.) | / | 0 |

softmax function to eliminate the comment content of irrelevant to usage scenes; (3) scene and feature lexicon formation layer, which obtains the vector expression of scene words and feature words from the filtered comment content with the help of Word2Vec, and forms the corresponding lexicon; (4) scenario-feature mining layer, which mines the co-occurrence relationship between scene and feature in the lexicon through the construction of co-occurrence matrix, and evaluates the association strength of the co-occurrence of scene and feature with mutual information.

## BERT-based classifier construction of scene sentence

Since online comments are unstructured language and heavily colloquialized, there are a lot of scene-independent sentences. Table 1 shows the comment sentences that contain scenes and do not contain any scenes.

Category "1" includes description of scenes, category "0" does not include any description of scenes. The comment sentence "动力说来就来, 没有迟钝, 超车自信很多" ("Power comes too soon, no any sluggishness, and there is confidence in overtaking in overtaking") contains the scenario of "超车" ("overtaking"), in which users mainly focus on the product feature of "power". However, the proportion of comments containing scenario-features in user comments is very low, and a large number of noisy sentences will seriously affect the performance of the usage scenario mining model. So it is necessary to eliminate scenario-irrelevant sentences in order to accurately identify users' expressions about product usage scenarios.

Bidirectional Encoder Representations from Transformers (BERT) (*Devlin et al., 2018*), released by Google AI in 2018, is a linguistic representation model developed based on deep learning, and significantly improves accuracy in several natural language processing tasks. The goal of BERT is to obtain a textual representation containing extremely rich semantic information by pre-training the model with a large-scale unlabeled *corpus*, and
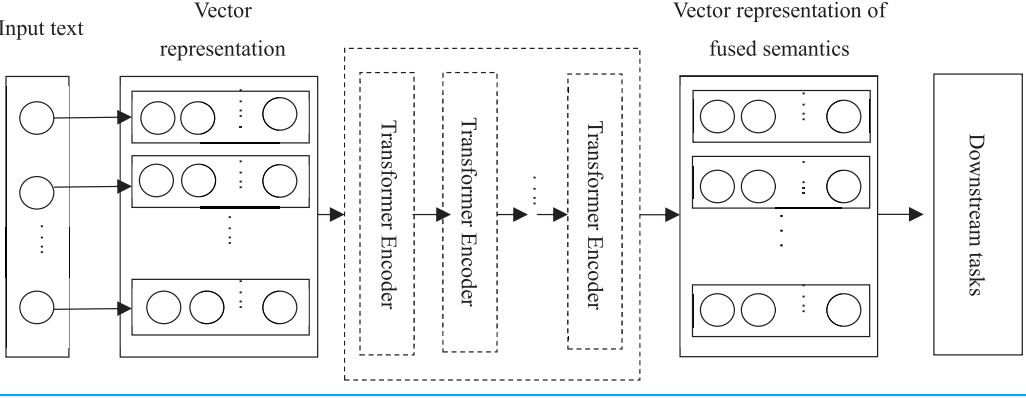

**Figure 2  BERT model structure (structure diagram).**

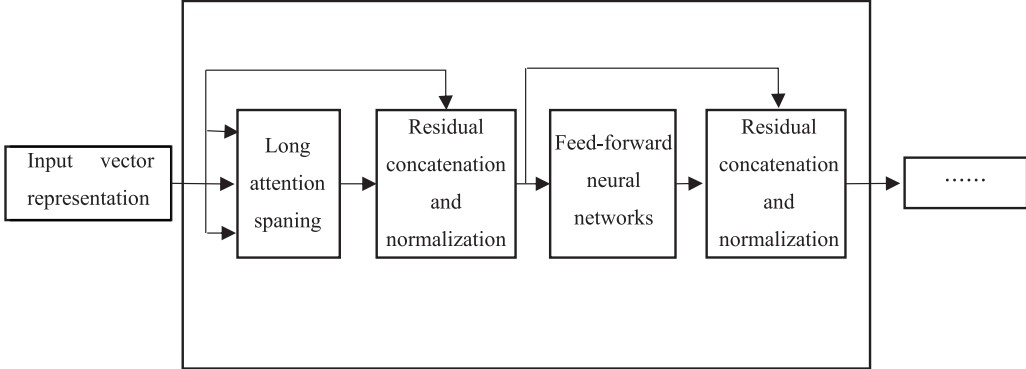

**Figure 3  Model structure of transformer encoder.**

then fine-tune different downstream tasks for the application of the textual representation. The network architecture of BERT uses a multilayer Transformer (*Vaswani et al., 2017*) structure, which improves the more intractable long-term dependency problem of traditional RNNs and CNNs. BERT is designed to pre-train deep bidirectional representations of unlabeled text through jointly conditioning the left and right contexts of all layers. It can fine-tune the pre-trained model in numerous text processing only using an additional output layer, and has shown its excellent results. For example, *Lu, Du & Nie (2020)* and *Croce, Castellucci & Basili (2020)* both obtained good text classification results based on the BERT model. In view of its excellent performance, based on BERT model, this article proposes a scene sentence classifier to accurately identify comment sentences containing scene descriptions with the minimized loss of effective information. Figure 2 shows the basic structure of the BERT model. Because BERT is a generic pre-trained model and needs to be fine-tuned to connect various downstream tasks, it is impossible to use a fixed decoding structure. It can extract various linguistic text features and realize the output of text representation only by stacking the coding part of Transformer.

The core of the BERT model lies in the adoption of the Transformer structure, in which each Transformer Encoder consists of an attention mechanism layer and a feedforward neural network layer, as shown in Fig. 3.

However, the weights are not shared among each other. There are residual connections and layer normalization after both layers, mainly to address gradient disappearance/ explosion as well as to accelerate convergence and training. Each word in the text is processed by word embedding and formed into a matrix, and then is input into the attention mechanism layer. The representation of each word, which consists of word vector, word position vector and text vector describing related semantics, captures the sequential information of the sequence. Firstly, the vector of each word is multiplied with a randomly initialized matrix to generate the three matrices of "*Query*", "*Key*" and "*Value*". The score of the corresponding word is calculated by $\frac{Query \cdot Key}{\sqrt{d_k}}$, and is input into the softmax function for normalization, which denotes the degree of representation of the current word in each word position in the text. In order to reduce the attention to irrelevant words, the output of the attention mechanism layer is obtained by accumulating the product of the normalized scores and the matrix "*Value*". Transformer Encoder adopts a multi-headed attention mechanism to extend the ability of focus on different locations, and outputs *g* matrices, which are concatenated and multiplied with a randomly initialized matrix. They are transformed into a single matrix by such a linear transformation. This matrix is directly added to the output of the attention mechanism layer. The residual is concatenated to avoid gradient problems, and then is normalized. The result is input into a fully connected feedforward neural network, which consists of two linear transformation layers. The ReLU function is used as the activation function because of its fast convergence and no gradient disappearance problem. The result after the residuals is concatenated and normalized is input into the next Transformer Encoder layer.

Many specific tasks can be embedded with pre-trained BERT representation layers, such as named entity recognition (*Souza, Nogueira & Lotufo, 2019*), sentiment analysis (*Sun, Huang & Qiu, 2019*), *etc*. The softmax function is often used to solve classification problems, and its function values are denoted by probabilistic sequences. BERT prefixes each input sentence with a "CLS" symbol when scene sentences are classified. This symbol does not have any textual semantic information when it is included in the output vector representation, which guarantees the fairness of the semantic information integration of the whole text. In this article, a softmax classifier is added to the right of BERT, *i.e.*, behind the output of Fig. 2, to achieve the recognition of scene comment sentence. After the calculation of softmax layer the output vector sequence of Fig. 2 can be transformed into different probability expressions which indicates the possibility of the comment text belonging to the category of scene sentences to find the maximum probability term, and complete the text classification task. Taking the comment sentence "动力说来就来, 没有迟钝, 超车自信很多" ("Power comes too soon, no any sluggishness, and there is confidence in overtaking in overtaking") as an example, the word vector, word position vector and text vector of each word of the text are together input into the model, and the rich semantic features of the text are extracted by stacking multi-layer Transformer Encoder. Then, it outputs the vector expression including the semantic information of the full text. The probability of the text belonging to the scene category "1" and "0" is gotten after this vector expression is input into softmax, which is used to determine whether the

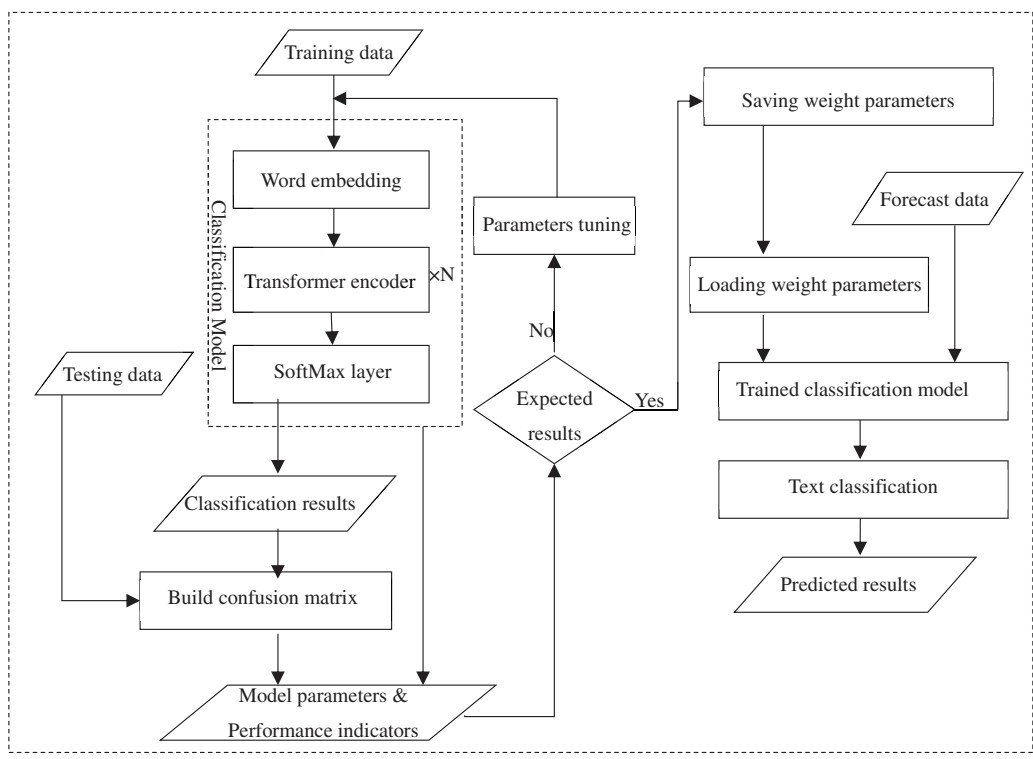

**Figure 4 Process structure of BERT-based scene sentence classifier.**

text should be rejected. The flow structure of the BERT-based scene sentence classifier is shown as Fig. 4.

In summary, a BERT-based classifier for scene sentence recognition from online reviews was constructed as follows. First, the classification model was trained using labelled data (training dataset) in which the Encoder structure of the stacked Transformer model was used to capture text features and a softmax layer was added to solve the classification problem (*Jiang et al., 2018*; *Du & Huang, 2018*). The softmax layer transforms the model output vector sequences into different probability expressions to identify sentences containing scene descriptions in the *corpus*. The classification results predicted by the model are compared with the test set to construct a confusion matrix to calculate the model performance metrics, and the corresponding parameters of the trained model are generated. The performance metrics are used to evaluate the performance of the model, and if the desired results are achieved, the trained weight parameters will be saved, otherwise the parameters of the model are adjusted and trained again using the grid search method until the optimal results achieved. In the case of classification prediction of new data, no further training is required and the weight parameters and the unlabeled statements are directly loaded into the trained classification model to get the classification results.

Precision, recall and F1 value are important metrics to evaluate the performance of classifiers on text classification (*Chinchor, 1992*). F1 value is the summed average of

precision and recall. These metrics are used to measure the effectiveness and error level of the proposed models. They are defined as follows:

$$Precision = \frac{N}{M} \tag{1}$$

$$Recall = \frac{N}{S} \tag{2}$$

$$F1 = \frac{2 * Precision * Recall}{Precision + Recall} \tag{3}$$

where N denotes the number of samples whose prediction results are correctly identified as positive categories, M denotes the number of all samples whose prediction results are positive categories, and S denotes the number of all samples that are positive categories in fact.

## Scene—feature mining model design

### Skip-gram-based word vector generation

The usage scenes and feature attributes pertaining to a product are limited and often repeatedly mentioned in users' reviews, and usage scenes and product features often exist in the form of verbs and nouns in users' reviews. Therefore, the nouns, noun phrases, verbs and verb phrases that repeatedly mentioned in the reviews can be used as candidate seed words for the scenario and feature lexicon, which are filtered by word frequency statistics method.

The lexicon is expanded to make the scene lexicon and feature lexicon more general based on the Word2Vec (*Mikolov et al., 2013a*). Word2Vec, a word quantization model proposed by *Mikolov et al. (2013b)*, is one of the most used methods for processing Chinese text vectors (*Lilleberg, Zhu & Zhang, 2015*). Word2Vec adopts a neural network approach to extract target information from unlabeled *corpus*. Word2Vec mainly has two models, named CBOW (Continuous Bag of-words) and Skip-gram, respectively. CBOW model predicts the probability of the current word based on contextual information, input the word vector of the contextual word of a feature word into it, and output the word vector of a specific word. Skip-gram model predicts the probability of the context based on the current word, input a specific word, and output the relevant word vector of the corresponding context of the specific word. Compared to CBOW model, Skip-gram model has better effectiveness on text processing and can better process low-frequency words (*Ma & Zhang, 2015*; *Fulin, Yihao & Xiaosheng, 2015*). Therefore, Skip-gram model is employed to train word vectors in this article, and the model structure is shown in Fig. 5.

Skip-gram model is mainly used to learn the weight correction parameters by back propagation and stochastic gradient descent algorithm, and get the optimized value of the objective function. Given the central word $v_i$, the contextual words $v_{i-t} \cdots v_{i-1}, v_{i+1} \cdots v_{i+t}$ within the window $t$ before and after $v_i$ are predicted. The objective function of Skip-gram model for training *corpus* $v_1, \cdots, v_r$ can be expressed as:

$$\mathbb{C} = \frac{1}{R}\sum_{r}^{R} \sum_{-i<f<i, f\neq 0} log_P\left(v_{\gamma+f}|v_{\gamma}\right) \tag{4}$$

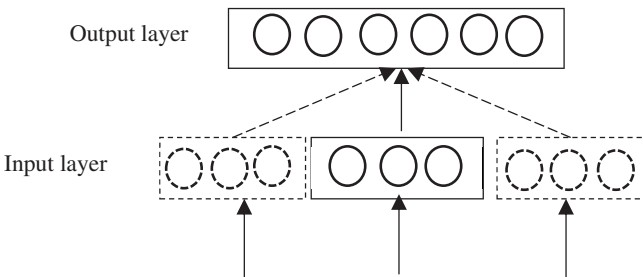

Output layer

Input layer

**Figure 5 Skip-gram model structure.**

where $R$ represents the number of words in the *corpus*, $v_\gamma$ denotes the central word, $v_f$ denotes the contextual word, $f$ is the window size, and the training objective is to maximize the objective function $\mathbb{C}$.

### Related lexicon expansion

The vector representation of each word as the central word and the context can be obtained through the training of the Skip-gram model, and the vector of the central word is selected as the final vector in this article. Then, this trained word vector model is used to calculate the semantic similarity between the words in the comment *corpus* and the seed word list, which is measured by the cosine similarity and is calculated as follows:

$$Cosine\ similarity = \frac{\vec{a} * \vec{b}}{|\vec{a}| * |\vec{b}|} \tag{5}$$

where $\vec{a}$ and $\vec{b}$ denote the word vectors of the seed words and the words in the *corpus*, respectively. The words with similarity greater than a threshold are selected as supplementary words in the *corpus*, and added to the scene lexicon and the feature lexicon.

### Scene—feature co-occurrence association

The co-occurrence matrix is constructed to represent the "many-to-many" associations between user scenarios and product features. Word co-occurrence in Chinese constitutes a complex co-occurrence network, which contains extremely rich semantic information that can be used in a variety of ways (*Garg & Kumar, 2018*; *Liang & Wang, 2019*). We mainly focus on the product features in a given usage scenario, and the word co-occurrence frequency is employed as the main indicator of association between scenes and features to measure their association strength. The other complex semantic information contained in co-occurrence networks are ignored. Multi-word co-occurrence, the association of multiple words in terms of semantic and other features, is determined based on statistical methods. In the *corpus*, these words that frequently co-occur with other words within a certain window size are correlated with each other on certain features, and the more frequently they co-occur, the stronger this correlation is (*Zhang et al., 2012*; *Figueiredo et al., 2011*). The construction procedures of scene-feature co-occurrence matrix is described as follows:

**Step 1:** Generate a two-dimensional empty matrix **T**. Construct the scene-feature co-occurrence matrix with all extracted feature words as the rows of the matrix and all scene words as the columns of the matrix.

**Step 2:** Assign values to the matrix. As can be seen from the processed product review contents, the text length is difficult to be unified because online reviews are a kind of spoken text, each opinion sentence has its own characteristics. Therefore, the size of the co-occurrence window is set as a dynamic value according to the semantic features of Chinese, *i.e.*, the length of the split sentence. The values of all co-occurring scene-feature word pairs are accumulated to form the final co-occurrence matrix $T = [S, F]$, as shown in Eq. (6).

$$T = \begin{matrix} S/F \\ F_1 \\ \cdots \\ F_i \\ \cdots \\ F_m \end{matrix} \begin{bmatrix} a_{11} & \cdots & a_{1j} & \cdots & a_{1n} \\ \cdots & \cdots & \cdots & \cdots & \cdots \\ a_{i1} & \cdots & a_{ij} & \cdots & a_{in} \\ \cdots & \cdots & \cdots & \cdots & \cdots \\ a_{m1} & \cdots & a_{mj} & \cdots & a_{mn} \end{bmatrix} \quad \begin{matrix} S_1 & \cdots & S_j & \cdots & S_n \end{matrix} \tag{6}$$

where $S_j$ denotes the set of scene words and $F_i$ denotes the set of corresponding feature words.

The word co-occurrence is widespread in natural language texts and implies a correlation relationship between words (*Bullinaria & Levy, 2007*). The strength of this co-occurrence relationship is generally measured by using mutual information (*Salle & Villavicencio, 2019*). Mutual information, a concept in information theory that measures the degree of signal association, has been extended to the text processing areas to measure the degree of association of topics with features (*Loeckx et al., 2009*). In this article, the mutual information $MI(f_i, s_j)$ is used to measure the degree of correlation between the scene word $S_j$ and the feature word $f_i$, which indicates the attention to a feature in a particular usage scenario. The mutual information is defined as shown in Eq. (7):

$$MI(f_i, s_j) = log_2 \left( \frac{P(f_i, s_j)}{P(f_i) * P(s_j)} \right) \tag{7}$$

where $P(f_i)$, $P(s_j)$, and $P(f_i, s_j)$ denote the probability of occurrence of the feature word $f_i$, the scene word $S_j$, and the probability of co-occurring within a certain window, respectively. When the value of $MI(f_i, s_j)$ is greater than zero, the larger the value the stronger the correlation between the scene word and the feature word, and *vice versa*. And when the value of $MI(f_i, s_j)$ is less than zero, it means the two words are not correlated.

It is highly susceptible to the dimensional explosion problem when the co-occurrence matrix $T = [S, F]$ is employed to represent the scene-feature association, but the proportion of valid co-occurrence word pairs is not too large in this study settings. Although expression errors can cause co-occurrence of few words, it occurs less. Therefore, those words can be filtered out by using of co-occurrence intensity evaluation method, and reduce the dimension of co-occurrence matrix. Based on the above results, an association

word library consisting of triads such as <scene word-feature word ($s - f$), number of co-occurrence ($t$), association intensity ($mi$)> is constructed for further analysis.

# EXPERIMENTS AND RESULTS

## Experimental data sources and pre-processing

The experimental data comes from the Chinese professional auto community. Pacific Auto (pcauto.com.cn), a professional automobile portal in China, provide netizens with the information about users' experiences and opinions on various brands of models. Based on user scale and market reputation, Volkswagen Land Rover and Nissan Xuan Yi were selected as the research objects. The reviews, which ended in March 31, 2021, were crawled by using Python programming method. A total of 10,927 Chinese reviews were identified as the valid data for analysis after clearing the reviews that are low readable, duplicative, and published by the same user. Through sentences tokenizing and de-duplicating, we obtained 48,694 comments for the Volkswagen Land Rover and 39,727 comments for the Nissan Xuan Yi. More than 10,000 reviews of Honda Fit, Buick Elite, Skoda Minolta and Dongfeng Peugeot 408 were also crawled, and were taken as training set and test set by manually labeling. Finally, 8,990 labeled data were obtained.

## Scene sentence acquisition

The efficient, convenient, and extendable characteristics of Tensorflow framework make it one of the most popular frameworks for deep learning, and it works better in data processing and transfer when combined with Numpy. Therefore, Tensorflow framework was adopted to build the BERT-based classifier. The manually labeled dataset is randomly divided into training and test sets by a ratio of 7:3. The training parameters are set as the maximum length of each sample (the maximum number of words), the longer the sample length, the more contextual information is involved. Thus, max_length is set to 32. And batch_size is set to 128. Learning_rate is set to 2e−5. The attention head, g, is set to 8. The training times of all samples in the training set, number_of_epochs, is set to 8. And the rest of the parameters are set to default values.

The classifier effectiveness is tested using the pre-divided test set, and the precision rate is 86.84%, the recall rate is 87.24%, and the F1 is 87.04%, which achieves a better classification performance. Accordingly, the scene comment sentences in the unstructured online comment set were well recognized, and a total of 15,132 scene comment sentences were obtained for the Volkswagen Ranger and 11,625 for the Nissan Hennessey. The NB algorithm is simple, fast and works well for text classification problems. Support vector machine (SVM) has better generalization capability and robustness for the noise in various linear or nonlinear classification problems. Feedforward neural network (FNN) models can get better performance in dealing with complex features and have higher adaptivity. Long short-term memory (LSTM) models have better memory capability. A recurrent neural networks (RNN) and FNN also have strong performance and a wide range of application scenarios. These methods have a wide range of applications and good performance in different fields, and are commonly used benchmark models, which are important for comparing the performance of different algorithms for the same problem.

**Table 2 Model performance comparison.**

| Algorithm model | Recall | Precision | F1 |
|---|---|---|---|
| NB | 81.99% | 81.98% | 81.98% |
| SVM | 85.84% | 85.84% | 85.84% |
| KNN | 68.64% | 77.26% | 68.64% |
| FNN | 82.27% | 82.53% | 82.40% |
| RNN | 85.53% | 82.70% | 84.10% |
| LSTM | 83.26% | 83.24% | 83.24% |
| BERT | 87.24% | 86.84% | 87.04% |

Therefore, in this article, we choose the representative methods, Naïve Bayes (NB), SVM, k-nearest neighbor (KNN), FNN, RNN and LSTM, as the baseline models. The results of the comparison experiments are shown in Table 2. By comparison, it can be seen that SVM, KNN, RNN and other models do not perform as well as the proposed model in the article in terms of relevant metrics, *i.e.*, the BERT-based scene sentence classifier has a more significant performance improvement than other learning algorithms.

## Scene-feature extraction and co-occurrence

Jieba is an easy-to-use word segmentation tool, and provides multiple available word segmentation modes. The customized domain dictionaries can be allowed to upload to correct the segmentation results, which makes the word segmentation results more accurate. The reviews involved many non-standard languages such as dialects and homophones, which will cause segmentation errors. Jieba is adopted to segmentate the words of scene sentences, and build a customized dictionary with domain vocabulary to correct the word segmentation results. Then, the part of speech is tagged, stop words are loaded, and the words unrelated to scenes, features, and preferences are removed. The final comment dataset for analysis gets formed, and word frequency is conducted. By comparing the lexicons constructed with top50, top100 and top150 words as the seed words, respectively, the top100 words are adopted as the seed words according to the ratio of irrelevant words. The adopted seed words are further filtered to remove irrelevant domain words, and then are divided into scene seed words and feature seed words. The word segment *corpus* is input into the Word2Vec module of the third-party Gensim library for training to get the word vector. The training parameter sg is set to 1, the word vector dimension size is set to 600, the maximum possible distance between the current word and the predicted word windows is set to 5, and other parameters are set as default values. Some results of the trained word vector are shown in Table 3.

The similarity function is used to calculate the similarity between all words and the seed words, and some words similar to the scene word "城市" ("urban area") are shown in Table 4.

The scene and feature seed word list are expanded to form the final scenario/feature word lexicon by using the semantic similarity method. A total of 162 scene words and 165 feature words are obtained, as shown in Table 5:

**Table 3 Part of word vectors.**

| Feature/scene | Word vectors | | | | |
|---|---|---|---|---|---|
| 空间 (Space) | −0.13084756 | −0.08914103 | 0.18648699 | 0.7289831 | ⋯ |
| 动力 (Power) | −0.82737607 | 0.08043833 | 0.64954895 | 0.02611112 | ⋯ |
| 内饰 (Interior) | −0.51031667 | −0.02091435 | 0.71320736 | 0.27824414 | ⋯ |
| 外观 (Appearance) | 0.01480259 | −0.11592811 | 0.66189367 | 0.6344568 | ⋯ |
| 转向 (Turn) | −0.9206333 | −0.02755794 | 0.07776708 | 1.3021126 | ⋯ |
| 减速带 (Reduction belt) | −0.33325452 | 0.13271216 | −0.7284181 | 0.688151 | ⋯ |
| 城市 (City) | −0.21520491 | −0.01623194 | 0.09567793 | 0.18351892 | ⋯ |
| 超车 (Overtaking) | −1.401604 | 0.14913893 | 0.22020623 | 0.2937419 | ⋯ |

**Table 4 Words similar to "城市" ("urban area").**

| Vocabulary | Similarity | Vocabulary | Similarity |
|---|---|---|---|
| 市内 (In-City) | 0.9186254143714905 | 堵车 (Traffic jam) | 0.8528324961662292 |
| 市里 (In the City) | 0.9179604053497314 | 走走停停 (Walking and stopping) | 0.8503051996231079 |
| 城市 (Urban) | 0.9089120626449585 | 城区 (City) | 0.8496943116188049 |
| 郊区 (Suburbs) | 0.8931871652603149 | 上下班 (Commuting) | 0.8436557650566101 |
| 城里 (In Town) | 0.8924171328544617 | 市郊 (Suburban) | 0.8432121276855469 |
| 县城 (County) | 0.8834766149520874 | 城市道路 (City roads) | 0.8259782791137695 |

To continue, the scenario-feature co-occurrence matrix is constructed with scene words as rows and feature words as columns to evaluate the association strength between scenes and features. The scene words and feature words are identified from all the comment texts containing scene descriptions based each tokenized comment. Then, the accumulated co-occurrence times of scene word $S_i$ and feature word $f_j$ are gotten and assigned to the corresponding positions in the co-occurrence matrix to form the final co-occurrence matrix $T$, as shown in Eq. (8). The mutual information values between scene words and feature words are calculated using Eq. (7) to evaluate their association strength.

$$
\begin{array}{c}
\text{Scenarios/features} \quad \text{Household} \quad \text{Climbing} \quad \cdots \quad \text{Start} \quad \text{Urbanarea} \\
\begin{array}{r}
\text{Power} \\
\text{Space} \\
\text{Fuelconsumption} \\
\cdots \\
\text{Interior} \\
\text{Valueformoney}
\end{array}
\begin{bmatrix}
689 & 84 & \cdots & 1836 & 248 \\
576 & 1 & \cdots & 29 & 34 \\
15 & 7 & \cdots & 14 & 1266 \\
\cdots & \cdots & \cdots & \cdots & \cdots \\
141 & 17 & \cdots & 32 & 43 \\
81 & 0 & \cdots & 5 & 0
\end{bmatrix}
\end{array}
\tag{8}
$$

A total of 4,867 scene-feature word pairs were obtained, and 3,124 valid association word pairs were obtained by filtering with association strength threshold. Based on the product descriptions and configuration overviews provided by manufacturers, the usage scenarios and features defined by manufacturers were analyzed, among which 15 usage scenario words, 27 product feature words and 20 scenario-feature association word pairs

**Table 5 Product scenario/feature lexicon.**

| Seed words | Expanded words |
|---|---|
| 家用 (Household) | 家庭, 家用车, 代步, 代步车, 日常, 家庭型, 五口之家, 居家, 家用轿车 (Family, family car, taking vehicle, transportation vehicles, daily, family-type, a family of five, private car) |
| 加速 (Accelerate) | 提速, 起步, 超车, 爬坡, 踩油门, 三档, 上桥, 二档, 换挡, 憋车, 加速性, 加油, 一档, 平地, 二挡 (Pick up speed, start, overtake, climbing, step on the gas, third gear, on the bridge, second gear, shift, hold the car, acceleration, refueling, first gear, flat, second gear (in wrong Chinese word)) |
| 市区 (Downtown) | 市内, 市里, 城市, 郊区, 城里, 县城, 堵车, 城区, 上下班, 市郊, 城市道路, 堵, 拥堵, 镇区, 乡, 镇, 不堵车, 塞车, 城, 上班, 快速路 (In city, in the city, suburban, city, county, traffic jam, urban, commuting, suburb, city roads, blocking, congestion, township, town, no traffic jam, traffic jam, city, go to work, expressway) |
| ... | ... |
| 外观 (Appearance) | 外形, 内部空间, 空间, 中排, 轴距, 左右间, 效率, 纵向, 紧凑型, 外型, 车座 (Shape, interior space, space, middle row, wheelbase, the left and right sides, efficiency, longitudinal, compact, appearance, car seat) |
| 动力 (Power) | 涡轮, 内饰, 输出, 排量, 加速度, 手动挡 (Turbo, interior, output, displacement, acceleration, manual transmission) |

**Table 6 Comparison between manufacturers' definition and model identification.**

| Methods | Scene words | Characteristic words | Effective scenario-featured associated word pairs |
|---|---|---|---|
| Model identification | 162 | 165 | 3,124 |
| Manufacturers' definition | 17 | 36 | 42 |

were obtained about Volkswagen Land Rover, 16 usage scenario words, 34 product feature words and 34 scenario-feature association word pairs were obtained about Nissan Xuan Yi. And a total of 17 usage scenario words, 36 product feature words and 42 scenario-feature association word pairs were obtained by fusing and de-duplicating. All the scenario words, such as "起步" ("start") and "转弯" ("turn"), and the feature words, such as "轮胎" ("tire") and "油耗" ("fuel consumption"), obtained from the manufacturers' definition had been recognized by our proposed model. Not only that, their synonyms, such as "加油" ("refueling"), "提速" ("speed"), "前座" ("front seat"), "后座" ("back seat"), "seat" ("座位"), are also recognized by our proposed model, which verifies that our proposed model can identify more scenarios and features than manufacturers' definition. The comparison between the identification results of proposed model and the manufacturer's definition is shown in Table 6.

Table 7 shows the association strength of fine-grained product features with the user scenario "提速" ("speed up"). As shown in Table 5, the features associated with the scenario "speed up" is "手动" ("manual"), "排量" ("displacement"), "油门" ("throttle") and "动力" ("power"). The association word library is formed by the triads such as <提速-手动 (Boost-Manual), 26, 1.401816>. According to this library, manufactures can effectively identify the product features that users most focus of in a specific scenario, improve them and optimize the configuration of product features to effectively meet the personalized requirements.

| Table 7 Product features associated with the "speed-up" scenario. | | |
|---|---|---|
| **Features** | **Number of co-occurrences** | **Strength of association** |
| 手动 (Manual) | 26 | 1.401816 |
| 做工 (Workmanship) | 2 | −0.76811 |
| 油门 (Throttle) | 97 | 1.790111 |
| ... | ... | ... |
| 后尾箱 (Rear trunk) | 0 | 0 |
| 动力 (Power) | 679 | 1.603128 |

As can be seen from Tables 6 and 7, the proposed framework can identify more scenarios and features than the manufacturers' definition in the product descriptions and configuration overviews. A lot of usage scenarios and related features that are of interest to consumers but not defined by the manufacturers are identified, and are more fine-grained.

A feature lexicon is constructed using word embedding techniques and other techniques in terms of the differences between product features and usage scene expressions, and the text features are captured by stacking the Transformer model Encoder structure (shown in Fig. 3). The classification problem is solved by using a softmax function to convert the model output vector sequences into different probabilistic expressions for the identification of the sentences containing scene descriptions. The *corpus* is used to construct a BERT-based classifier for scene sentence recognition from online reviews (as shown in Fig. 2). Then the model is trained using the annotated dataset, and the optimal model is saved, which is loaded in the experiment to predict the unannotated data and achieve the classification of the target text (the process is shown in Fig. 4). Finally, the effectiveness of the proposed process framework is verified through experiments with user comment data in an auto community. The comparison results with classification results of other models and manufactures' definition shows the superior performance of this process framework. The process framework also can provide the strength of association between different features and scenarios, enabling manufacturers to quickly identify more fine-grained usage scenarios and their relevant product features in practice, providing guidance for product improvement.

Although there is a wide variety of products, the reviews for different products are much the same. For example, the review of a cell phones,"这款手机的摄像头能拍出非常好的照片, 可与相机的相媲美 (The camera of this phone takes very good pictures, comparable to the pictures taken with camera.)", is similar to the review of a car, "车辆空间非常大, 回家或露营都非常合适 (The vehicle is very spacious and is great for driving home or camping.)". These two reviews of the two different products both can be analyzed by the same method this article constructed, involving the scenarios of "taking pictures" and "going home or camping" respectively, with their corresponding features classified as "camera" and "space". Thus, online reviews of other kinds of products also can be analyzed by using the proposed process framework, which further confirms the generalization capability of this method.

## CONCLUSION

Data has increasingly become important strategic resource in big data era. Online review, which is associated between users and products, is an important source of information about user preference and usage experience. With the advent of the scenario era, user pursuits not only the meet of functions but also the cooler experience in a given scenarios. Thus, the product with the configurations according to user's specific scenario meets requirement more effectively. In this article, we proposed a scenario-feature mining framework from online reviews. First, a text classifier is constructed based on BERT model to filter out scenario-irrelevant comments. Second, the usage scenario and feature lexicon are constructed based on Skip-gram model. Finally, a scenario-feature co-occurrence matrix is constructed to mine scenario-feature pairs, and scenario-feature associations at a fine-grained level are achieved by using mutual information. To verify the proposed framework, the reviews about Volkswagen Land Rover and Nissan Hennessy are selected as the cases for experiment. The results of the experiment show that this framework can effectively identify the specific usage scenarios and the corresponding product features, which provide a strong basis for improving product design based on user needs and improve manufacturer's competitiveness. Nevertheless, there are still some shortcomings in this study in terms of sentiment preference and classifier training for the association between usage scenario and product feature. The unsupervised algorithm with a lower cost and sentiment analysis will be further introduced into the association construction between scenarios and features in the future.

### Funding
This study was funded by the National Office for Philosophy and Social Sciences (CN) (Grant No. 20BGL044). The funders had no role in study design, data collection and analysis, decision to publish, or preparation of the manuscript.

### Grant Disclosures
The following grant information was disclosed by the authors:
National Office for Philosophy and Social Sciences (CN): 20BGL044.

### Competing Interests
The authors declare that they have no competing interests.

### Author Contributions
- Xunjiang Huang conceived and designed the experiments, performed the experiments, analyzed the data, prepared figures and/or tables, authored or reviewed drafts of the article, and approved the final draft.
- Kang Yan conceived and designed the experiments, performed the experiments, analyzed the data, performed the computation work, prepared figures and/or tables, and approved the final draft.

## Data Availability

The code and raw data are available in the Supplemental File.

## Supplemental Information

Supplemental information for this article can be found online at http://dx.doi.org/10.7717/peerj-cs.1398#supplemental-information.

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
