# Peer review of "Scenario-feature identification from online reviews based on BERT"

_PeerJ Computer Science, doi:10.7717/peerj-cs.1398_

## Round 0.1 · original submission · Major Revisions

The manuscript needs a thorough proofread. The reviewers' comments showed the main weakness of the contribution is in the current presentation. The demonstration and validation of the methodology are not clear - therefore a major revision is needed.

Reviewer 1 ·

Basic reporting

The manuscript needs heavy proofreading
The literature is insufficient

Experimental design

The methods does not include any semantic or language specific information.
The methods must understand the dynamics of theoretical aspects of word network structure through literature:
1. Liang, W., & Wang, K. (2019). Relationships among the statistical parameters in evolving modern Chinese linguistic co-occurrence networks. Physica A: Statistical Mechanics and its Applications, 524, 532-539.
2. Garg, M., & Kumar, M. (2018). The structure of word co-occurrence network for microblogs. Physica A: Statistical Mechanics and its Applications, 512, 698-720.

Validity of the findings

Experimental results should be expanded with more experimental analysis such as ablation study and error analysis.

Reviewer 2 ·

Basic reporting

The paper presents a scenario-feature identification method for discovering scenarios, features and their relationship from online reviews.

1. The organization of the paper is good and the writing is also clear.
2. The authors reviewed and summarized relevant literature reference.
3. The raw data is shared but the data is the tables is not complete. If the tables are too long, the authors can provide an excel file with all the complete tables in the supplementary document. The authors need to provide complete lists of identified scenarios, features, and their association pairs.

Experimental design

4. The research topic is very interesting to the audiences of the journal.
5. The development of a classifier based on BERT to filter scenario is not clear enough. The authors need to clarify how to create an annotated corpus to fine-tune the BERT model. Is the annotated corpus good enough to develop a robust classifier?
6. The workflow is not optimal. It would better to select scenario and feature keywords based on word embedding and semantic similarity and then develop an annotated corpus to fine-tune the BERT model.
7. Why not develop another classier to filter sentences with features?

Validity of the findings

7. The scenario-feature identification method is somewhat novel, which is similar to topic modeling methods to identify topic-keyword association.
8. Although the proposed method is used for mining scenarios, features, and their associations from online reviews, it is not clear how to validate the proposed method overall.

---

## Round 0.2 · Major Revisions

I have taken over the handling of this submission due to unavailability of the original academic editor. Given that the two original reviewers were not available, I have sent your submission to review to a new reviewer.

As you can see, the reviewer is supportive of your work, but does raise some concerns which would need to be addressed in a major revision. These include, among others, providing more up-to-date references of related work, as well as fleshing out some of the decisions made in your study design and the analysis.

Reviewer 3 ·

Basic reporting

The topic is very interesting. The manuscript is written very clear. However, some clarifications are needed.

1. The references in the introduction should be recent to justify more of the current work.
2. In introduction, line 49,50,51. Need proper justification with recent works.

Experimental design

The methodology is clearly explained.

Validity of the findings

1. The authors mentioned disadvantages such as subjectivity, high cost, coarse granularity, and limited scenario in the existing approaches. However, they did not mention anything about their proposed approach that how they overcame these disadvantages.

2. The comparison should me more focused between the deep learning approaches rather than the traditional machine learning approaches.

3. Need proper justification for selecting the methods for comparison.

---

## Round 0.3 · accepted · Accept

Following the last round of major revisions, the previous reviewer and me agree that the paper can now be accepted for publication in its present form.

Reviewer 2 ·

Basic reporting

I don't see the response to my comments. Thank you!

Experimental design

I don't see the response to my comments. Thank you!

Validity of the findings

I don't see the response to my comments. Thank you!

Reviewer 3 ·

Basic reporting

The authors have addressed all the concerned. The manuscript now can be accepted for publication.

Experimental design

N/A

Validity of the findings

N/A

Additional comments

N/A